# Facet-Dependent SERS Activity of Co_3_O_4_

**DOI:** 10.3390/ijms232415930

**Published:** 2022-12-14

**Authors:** Yibo Feng, Jiaxing Wang, Jixiang Hou, Xu Zhang, Yuhang Gao, Kaiwen Wang

**Affiliations:** Beijing Key Laboratory and Institute of Microstructure and Property of Advanced Materials, Faculty of Materials and Manufacturing, Beijing University of Technology, Beijing 100124, China

**Keywords:** SERS, Co_3_O_4_, facet-dependent, DFT simulation

## Abstract

Surface-enhanced Raman spectroscopy (SERS) is an ultra-sensitive and rapid technique that is able to significantly enhance the Raman signals of analytes absorbed on functional substrates by orders of magnitude. Recently, semiconductor-based SERS substrates have shown rapid progress due to their great cost-effectiveness, stability and biocompatibility. In this work, three types of faceted Co_3_O_4_ microcrystals with dominantly exposed {100} facets, {111} facets and co-exposed {100}-{111} facets (denoted as C-100, C-111 and C-both, respectively) are utilized as SERS substrates to detect the rhodamine 6G (R6G) molecule and nucleic acids (adenine and cytosine). C-100 exhibited the highest SERS sensitivity among these samples, and the lowest detection limits (LODs) to R6G and adenine can reach 10^−7^ M. First-principles density functional theory (DFT) simulations further unveiled a stronger photoinduced charge transfer (PICT) in C-100 than in C-111. This work provides new insights into the facet-dependent SERS for semiconductor materials.

## 1. Introduction

Surface-enhanced Raman scattering (SERS) spectroscopy has attracted lots of attention in broad fields such as food safety, chemical and biological sensors and trace detection [1,2,3,4,5,6]. Discovered in 1970s by Fleischmann et al., molecular adsorption onto a rough silver surface can lead to drastically increased Raman signal intensity [7]. Currently, it is generally accepted that the enhancement mechanism includes two main aspects. The primary one is electromagnetic enhancement (EM), suggesting that the electric field can be magnified when excitation takes place in noble and coinage metals (Au, Ag and Cu) with a strong localized surface plasmon resonance (LSPR) effect [8]. By EM, enhancement factors (EFs) up to the order of 10^6^ can be achieved. The other aspect is chemical enhancement (CM) that always occurs in a semiconductor or other plasmonic-free materials. In CM theory, photoinduced charge transfer (PICT) between a substrate and adsorbed molecule could enhance the vibrational scattering of adsorbates and obtain EFs of 10–100 [9,10,11,12,13,14]. Despite highly sensitive substrate noble metals being reported, the high cost and poor reproducibility and biocompatibility severely impede their practical applications. Recently, considerable research has focused on semiconductor-based SERS substrates. Compared to noble and coinage metals, semiconductors possess merits of controllable physical and chemical properties, a tunable morphology, great biocompatibility and low cost, which enable them to be promising candidates for SERS applications [15,16,17,18].

Semiconductor-based SERS mainly stems from the photoinduced charge transfer (PICT) between the substrate and adsorbed molecules [12,15,19,20]. According to Fermi’s golden rule:$$w_{lk}=\frac{2\pi }{\hbar }{\left|{H^{\prime }}_{kl}\right|}^{2}g\left(E_{k}\right)$$
where $w_{lk}$ is the electron transition probability, ${H^{\prime }}_{kl}$ is the matrix element of the highest-occupied molecular orbital (HOMO)–lowest-unoccupied molecular orbital (LUMO) transition, and $g\left(E_{k}\right)$ is the density of states (DOS); there is a positive correlation between the DOS and electron transition probability during the charge-transfer process. Consequently, the physical and chemical properties of the semiconductor surface and its interaction with the analyte molecule are crucial to the SERS performance of semiconductors [21,22]. Up to now, several strategies have been reported to enhance the charge-transfer efficiency, such as defect engineering, element doping and constructing two-dimensional (2D) semiconductors [23,24,25,26,27,28,29]. In conclusion, by engineering the property or density of surface adsorption sites and the energy band of materials to modulate substrate–adsorbate interaction, the charge-transfer process can be adjusted. Although outstanding SERS activity has been achieved via these strategies, novel methods are still highly desired to further improve the performance of semiconductor-based SERS sensors.

Since PICT takes place on the surface of semiconductors, SERS activity of the semiconductor is inevitably affected by the surface atomic configurations. Surface atomic arrangement and the local coordination of active sites that adsorb analyte molecules determine the adsorption energy and charge-transfer process fundamentally. As a result, facet engineering emerges as an effective strategy to modify the CT process as the surface atom configurations are closely related to exposed crystal facets [14,30,31,32]. For example, Guo et al. demonstrated that the {100} faceted Cu_2_O microcrystals had the highest charge-transfer efficiency and exhibited a better SERS enhancement of the 4-nitrobenzenethiol (4-NBT) molecule than {110} and {111} faceted Cu_2_O [33]. Luo et al. demonstrated that the {0001} and {10$\overset{-}{1}$1} facets of ZnO exhibited superior SERS performance for 4–NBT compared with the {10$\overset{-}{1}$0} facet for their promoted charge-transfer process between the ZnO substrate and 4–NBT molecule [34].

In this work, high-uniform facet-defined (dominantly exposed {100}, {111} facets and co-exposed {100}-{111}) Co_3_O_4_ microcrystals (denoted as C-100, C-111 and C-both, respectively) were successfully prepared via a facile hydrothermal method with cube, octahedron and truncated octahedron shapes, respectively. These model crystals were applied as SERS substrates to detect R6G, adenine and cytosine. The lowest detection limits (LODs) to R6G and adenine can reach 10^−7^ M for {100} faceted Co_3_O_4_ (C-100), surpassing that of C-111 and C-both by at least one order of magnitude. First-principles calculations further revealed that {100} facets can transfer more electrons from adenine to Co_3_O_4_, promoting the CT process and increasing the molecular polarization.

## 2. Results and Discussion

### 2.1. Characterization of Faceted Co_3_O_4_ Microcrystals

To study the facet-dependent SERS activity, well-defined Co_3_O_4_ microcrystals with different exposed facets were synthesized following a hydrothermal–calcination two step method according to the previously reported literature [35]. The details of all synthesis routes are listed in the Materials and Methods section. The morphology of these microcrystals was characterized by scanning electron microscopy (SEM), as shown in Figure 1. Nanostructures with a cube shape for C-100, a truncated octahedron shape for C-both and an octahedron shape for C-111 were observed. The three samples are all uniformly distributed with sizes near 1 µm, and it is noted that cubic C-100 is slightly smaller than the others (~300 nm). High-angle annular dark-field scanning transmission electron microscopy (HAADF-STEM) combined with energy-dispersive spectroscopy (EDS) mapping were used to further demonstrate the structure and elemental composition of these faceted Co_3_O_4_ microcrystals (Figure 2). As well as the confirmed regular shaped nanostructures, homogeneous distribution of Co and O were observed. X-ray diffraction (XRD) was utilized for the phase analysis of faceted Co_3_O_4_ (Appendix A). Several strong peaks located at 2θ = 19.0°, 31.3°, 36.8°, 44.8°, 59.4° and 65.2°, can be indexed to the diffraction planes of (111), (220), (311), (400), (511) and (440) of cubic Fd$\overset{-}{3}$m Co_3_O_4_ (JCPDS file no. 42-1467), indicating high crystallinity. In addition, there were no impurity peaks found in any samples. X-ray photoelectron spectroscopy (XPS) was further performed to investigate the valence states of the cobalt element and valence band maximum positions (Appendix A). As shown in Appendix A, the XPS spectra of Co 2p exhibited doublet peaks at around 780.0 eV and 795.0 eV, corresponding to Co 2p3/2 and Co 2p1/2, respectively. Moreover, the peak of Co 2p3/2 can be fitted with two peaks at 779.8 eV and 780.8 eV, indicating the co-existence of Co^2+^ and Co^3+^ [36]. The optical properties of faceted Co_3_O_4_ microcrystals were studied using ultraviolet–visible absorption spectroscopy (UV-vis). It can be seen from Appendix A that two broad absorbance peaks were observed between 500–600 nm and after 800 nm in all three samples, agreeing well with the previous reports [37]. No obvious strong plasmonic peaks can be observed, suggesting the lack of LSPR in the Co_3_O_4_ semiconductor. Combining the XPS valence band spectra (Appendix A) and the UV-vis spectra (Appendix A), the optical band structure can be derived. Owing to the almost identical data for three samples, it is reasonable that the optical band structures differ slightly in these different faceted Co_3_O_4_ samples.

### 2.2. SERS Activities of Faceted Co_3_O_4_ Microcrystals

To assess the SERS performance of faceted Co_3_O_4_ microcrystals, R6G was first used as a probe molecule. Appendix A showed the standard Raman signals of R6G powder. The prominent characteristic Raman peak at 612 cm^−1^ under excitation at 532 nm is ascribed to C–C–C in-plane ring vibration (Figure 3a), which is consistent with the previous literature [38,39]. Various R6G solutions with different concentrations (from 10^−4^ M to 10^−7^ M) were detected to investigate the SERS activity of different SERS substrates, i.e., C-100 (Figure 3b), C-both (Figure 3c) and C-111 (Figure 3d). For all substrates, the specific Raman signal intensity of R6G decreased with the decreasing solution concentration. When the concentration decreased to 10^−7^ M (100 nmol/L), only C-100 exhibited the characteristic peaks of the R6G. To validate the superiority of C-100 SERS activity, the SERS spectra of the three faceted Co_3_O_4_ were compared under the same R6G concentration (10^−4^ M). As shown in Figure 4e, the intensity of peak signals followed the order: C-100 > C-both > C111, demonstrating a higher performance of C-100 over C-111.

Adenine is an important DNA nucleic acid base involved in the storage and expression of genes. Diseases such as HIV infection, Down’s syndrome and cancer in the human body always occur with abnormal changes in the concentration of nucleic acid bases [40,41]. Therefore, we further extended the target molecule into adenine. The SERS spectrum of adenine powder was recorded using a 532 nm excitation laser (Appendix A). The prominent characteristic Raman peak at 722 cm^−1^ is ascribed to C–H ring breathing mode (Figure 4a), which is consistent with the previous literature [6,42]. Similarly, the SERS spectra of different adenine solution concentrations (from 10^−4^ M to 10^−7^ M) were obtained on these substrates (Figure 4b–d). The C-100 also exhibited the lowest detection limit of adenine, as low as 10^−7^ M. Figure 4e also demonstrates the higher activity of C-100 compared to C-both and C111, while the activity of C-both is between the other two substrates with solely exposed facets. This facet-dependent SERS activity phenomenon was further validated by the detection of cytosine. Systematic performance experiments were designed and carried out (Figure 5). As was expected, C-100 performed best again. The above results evidently indicated that the exposed facets of the semiconductor significantly determined its SERS sensitivity.

### 2.3. SERS Reproducibility and Stability of Faceted Co_3_O_4_ Microcrystals

Homogeneous SERS signals on a substrate are crucial to the practical application in the real world. To evaluate the reproducibility of C-100, 20 measurement points within an area of 2 × 2 cm^2^ were randomly selected to obtain spectra of different target molecules. As shown in Figure 6a–c, few differences could be found in the spectra curves, indicating that the C-100 SERS substrate has excellent reproducibility for the detection of R6G, adenine and cytosine. Characteristic peak intensities at 612, 722 and 790 cm^−1^ were chosen to assess the relative standard deviation (RSD) values. Finally, RSD values were calculated to be 6.0, 4.5 and 4.9% for R6G, adenine and cytosine, respectively. The long-term stability of C-100 was tested by comparing fresh substrates and substrates stored for three months (Appendix A). The results demonstrated that 84.9% of activity was maintained after storage. Therefore, the {100} facets-exposed Co_3_O_4_ microcrystals can be potentially served as a SERS monitor for biomolecule detection without any pretreatment or labeling process.

### 2.4. SERS Mechanism of Faceted Co_3_O_4_ Microcrystals

Density functional theory (DFT) calculations were employed to calculate the charge transfer and the adsorption behavior of the target molecules. The detailed original data was provided in Appendix A. Co_3_O_4_-(100) and Co_3_O_4_-(111) models were established and investigated as substrates to adsorb adenine (Figure 7a,d). The charge density difference results of the adenine–Co_3_O_4_ models revealed the electron transfer from adenine to Co_3_O_4_ [43] (Figure 7b,e). The Bader charge analysis and adsorption energy calculations were employed to semi-qualitatively describe the charge-transfer process and the interaction between the adsorbed molecule and the substrate [26,44]. C-100 (−2.07 eV) exhibited a much stronger interaction with adenine than C-111 (−0.85 eV). Moreover, C-100 can transfer more electrons (0.613 e) from the substrate to adenine (0.482 e) (Figure 7c,f). Therefore, the facilitated CT process on {100} facets result in enhanced CM and thus a higher SERS activity than can be achieved on {111} facets.

## 3. Materials and Methods

### 3.1. Materials

Sodium hydroxide (NaOH) pellets (ACS reagent, ≥97.0%), oxalic acid (H_2_C_2_O_4_, 98%) powder, cobaltous nitrate hexahydrate (Co(NO₃)₂·6H₂O, reagent grade, 98%) flakes were purchased from Sigma-Aldrich (St. Louis, MO, USA). Anhydrous alcohol (99%) was purchased from Sinopharm Chemical Reagent Co., Ltd., Shanghai, China. Ultrapure water (18.2 MΩ cm) was purified on a Milli−Q Advantage A10 (Millipore, USA). All reagents were used without further purification.

### 3.2. Synthesis of Facet Co_3_O_4_ Microcrystals

All the precursors were synthesized using a typical hydrothermal method [35]. To prepare for C-100 precursor, 0.01 mol of NaOH and 0.04 mol of Co(NO_3_)_2_ were stirred into 40 mL of deionized water, the resulting solution was transferred into a 100 mL Teflon-lined stainless steel autoclave and heated at 180 °C for 5 h. For C-111 precursor, 0.015 mol of H_2_C_2_O_4_, 0.05 mol of NaOH and 0.085 mol of Co(NO_3_)_2_ were mixed into 70 mL of deionized water and heated in an autoclave at 220 °C for 20 h. Similarly, 0.015 mol of Na_2_C_2_O_4_, 0.02 mol of NaOH and 0.085 mol of Co(NO_3_)_2_ were mixed into deionized water and heated in an autoclave to synthesize C-both precursor. After cooling to room temperature naturally, all the precursors were centrifuged and washed several times, and then dried at 60 °C for 5 h. Finally, the obtained solids were calcined at 500 °C in a muffle furnace for 3 h with a heating rate of 5 °C min^−1^.

### 3.3. Characterization

HR-TEM, STEM-HAADF and EDX elemental mapping characterization were performed by a Titan Themis G2 transmission electron microscope operated at 300 kV and equipped with a probe spherical aberration corrector. The binding energies obtained in the XPS spectral analysis were corrected for specimen charging by referencing C 1s to 284.8 eV. The UV-vis absorbance of the materials was measured on a quartz cuvette with wavelengths from 200 to 800 nm. X-ray diffraction (XRD) patterns were acquired with a Digaku D/max-2500 X-ray diffractometer. Scanning electron microscopy (SEM) images were obtained using a JSM-6360LA SEM at an accelerating voltage of 20 kV.

### 3.4. SERS Measurements

To study the SERS properties of these samples, a confocal microscopy Raman spectrometer (inVia^TM^ confocal Raman microscope, Renishaw, UK) was used as the measuring instrument. In all SERS tests, unless specifically stated, the excitation wavelength was 532 nm, laser power was 10 mW and the specification of the objective was 50 times long-focus objective lens (×50 L). A series of standard solutions of rhodamine 6G (R6G), adenine (A), cytosine (C) with different concentrations were used as the target molecules. Synthesized Co_3_O_4_ and 20 μL target molecules with different concentrations were dropped onto 5 mm × 5 mm polished silicon wafers. The mixture was dried at 60 °C.

### 3.5. Enhancement Factor Calculation

To calculate the enhancement factor (EF) of Co_3_O_4_, the ratio of SERS to normal Raman spectra (NRS) of rhodamine 6G (R6G), adenine (A) and cytosine (C) were determined by using the following calculating formula:$$\mathrm{EF}=\left(I_{SERS}/N_{SERS}\right)\;/\;\left(I_{NRS}/N_{NRS}\right)\;N_{SERS}=CVN_{A}S_{Raman}/S_{sub}\;N_{NRS}=M\rho hS_{Raman}N_{A}$$
where *I_SERS_* and *I_NRS_* refer to the peak intensities of the SERS spectrum and NRS, respectively. *N_SERS_* and *N_NRS_* refer to the number of R6G molecules in the SERS substrate and normal Raman sample, respectively. The radius of laser spot was approximately 0.5 µm, depth of laser penetration was about 1 µm. The data for R6G (bulk) on bare Si substrate were used as non-SERS-active reference. Specifically, the intensity was obtained by taking average from measurements of 20 spots, and *I_NRS_* was tested to be 6027 counts. *N_SERS_* was determined from the laser spot size illuminating the sample and the density of R6G molecules adsorbed on the Co_3_O_4_ chip surface. *C* is the molar concentration of the analyte solution. *V* is the volume of the droplet. *N_A_* is Avogadro constant. *S_Raman_* is the laser spot area (0.5 μm in radius) of Raman scanning. An amount of 20 μL of the droplet on the substrate was spread into a square of about 5 mm in side length after solvent evaporation, from which the effective area of the substrate, *S_Sub_,* can be obtained. Regarding R6G, the density (*ρ*) of the R6G solid was approximately 1.15 g/cm^3^, the confocal depth (*h*) of the laser beam was 1 μm and according to molecular weight (M), *N_NRS_* was estimated to be 5 × 10^10^.

In the SERS measurements, the Raman scattering peak at 612 cm^−1^ was selected for the calculation of the EF. Substituting these values of the above variable into the equation, the maximum EF could be concluded to be approximately 1.12 × 10^4^, while the concentration of R6G solution *C* = 10^−6^ M and the Raman intensity *I_SERS_* = 512 counts. The EF of adenine (5.81 × 10^3^) was also calculated following this procedure and the Raman scattering peak at 722 cm^−1^ was selected for the calculation of the EF. *I_NRS_* was tested to be 30,412 counts and the Raman intensity *I_SERS_* = 1336 counts. The EF of cytosine (2.78 × 10^3^) was also calculated. The Raman scattering peak at 792 cm^−1^ was selected for the calculation of the EF. *I_NRS_* was tested to be 15,012 counts and the Raman intensity *I_SERS_* = 315 counts.

### 3.6. DFT Calculations

The density functional theory (DFT) calculation was performed in “Vienna ab initio simulation package” (VASP5.4.1) with the generalized gradient approximation (GGA) using Perdew−Burke−Ernzerhof (PBE) [6,45,46]. DFT-D3 was adopted to correct for the van der Waals interaction. [47] A plane-wave basis set with cut-off energy at 400 eV and the projector-augmented wave method framework were adopted. The Gaussian smearing width was set to 0.2 eV. 1 × 1 × 1 K points were set in the Brillouin zone for all models. As previously reported [48], we used U = 5.9 eV for the 3D states of all Co atoms in our models. The bottom six layers of the C-100 and ten layers of the C-111 slabs were fixed during the relaxation, whereas the top two layers were fully relaxed until the energy and force converged within 10^−4^ eV and 0.02 eV Å^−1^, respectively. The models with 448 atoms were used for C-100 and C-111, and the surface configurations were determined based on surface energy calculations. The vacuum space was set to 20 Å. The adsorption energy (E_ads_) of adenine molecule was defined as [49]:E_ads_ = E_total_ − E_sub_ − E_molecule_
where E_ads_ is the adsorption energy of adenine on Co_3_O_4_, and E_total_, E_sub_ and E_molecule_ are the total surface energy, clean surface energy and molecule energy, respectively.

## 4. Conclusions

In summary, we demonstrated a facet-dependent SERS phenomenon by employing three types of faceted Co_3_O_4_ microcrystals with dominantly exposed {100}, {111} facets and co-exposed {100}-{111} facets as SERS substrates to detect R6G and adenine. A higher SERS performance on {100}-Co_3_O_4_ was obtained compared with {111} facets and co-exposed facets Co_3_O_4_. The lowest detection limits for R6G and adenine, as low as 10^−7^ M, were achieved with enhancement factors of 1.12 × 10^4^ and 5.81 × 10^3^, respectively. Meanwhile, the Co_3_O_4_ microcrystals exhibited good SERS reproducibility (RSDs less than 6%) and stability (more than three months). Density functional theory calculations were conducted to explore the charge-transfer process in different facets of Co_3_O_4_ and the adsorbed adenine molecules. The calculated results verified that the active sites on {100} facets can transfer more electrons from adenine to Co_3_O_4_ and have stronger adsorption interaction with adenine compared with {111} facets. This work may advance insights into the correlation between the structure and property of a semiconductor for SERS applications.

## Figures and Tables

**Figure 1 ijms-23-15930-f001:**
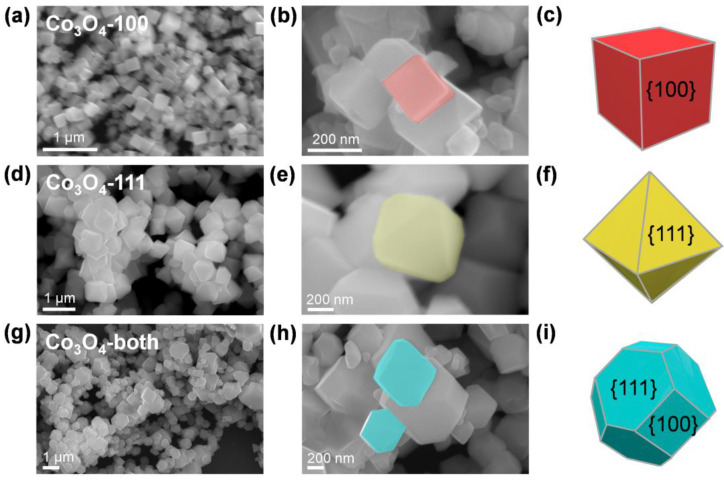
Microstructures of different faceted Co_3_O_4_ samples. (**a**,**b**) SEM images and (**c**) schematic illustration of C-100. (**d**,**e**) SEM images and (**f**) schematic illustration of C-111. (**g**,**h**) SEM images and (**i**) schematic illustration of C-both.

**Figure 2 ijms-23-15930-f002:**
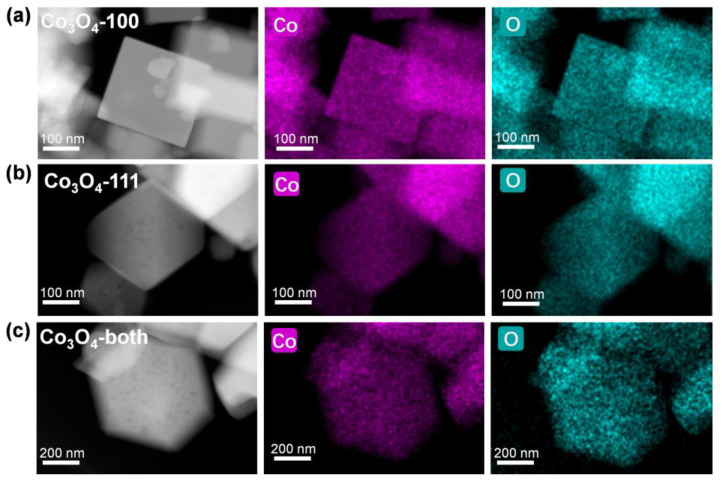
High-angle annular dark-field scanning transmission electron microscopy (HAADF-STEM) and energy-dispersive spectroscopy (EDS) mapping images of faceted Co_3_O_4_ samples. (**a**) C-100; (**b**) C-111; (**c**) C-both.

**Figure 3 ijms-23-15930-f003:**
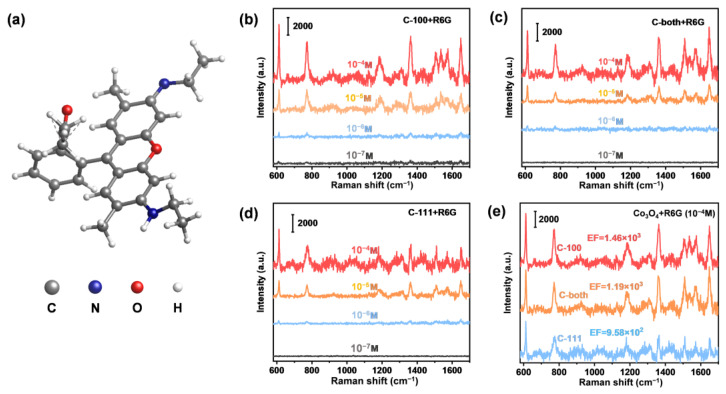
SERS spectra of R6G adsorbed on faceted Co_3_O_4_ samples. (**a**) Molecule structure of R6G; (**b**) SERS spectra of R6G adsorbed on C-100 within a concentration range from 10^−4^ to 10^−7^ M; (**c**) SERS spectra of R6G adsorbed on C-both within a concentration range from 10^−4^ to 10^−7^ M; (**d**) SERS spectra of R6G adsorbed on C-111 within a concentration range from 10^−4^ to 10^−7^ M; (**e**) SERS spectra of R6G adsorbed on faceted Co_3_O_4_ samples at 10^−4^ M.

**Figure 4 ijms-23-15930-f004:**
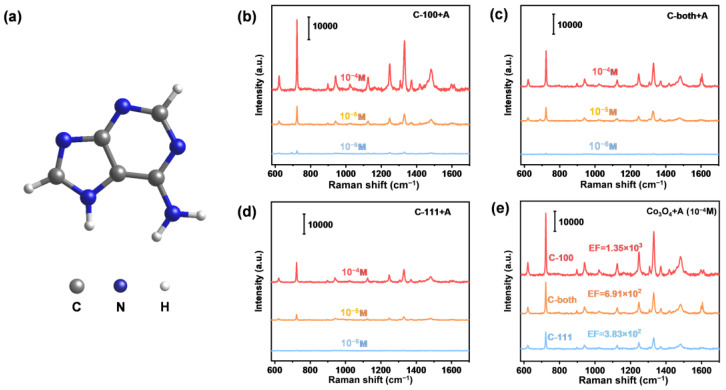
SERS spectra of adenine adsorbed on faceted Co_3_O_4_ samples. (**a**) Molecule structure of adenine; (**b**) SERS spectra of adenine adsorbed on C-100 within a concentration range from 10^−4^ to 10^−6^ M; (**c**) SERS spectra of adenine adsorbed on C-both within a concentration range from 10^−4^ to 10^−6^ M; (**d**) SERS spectra of adenine adsorbed on C-111 within a concentration range from 10^−4^ to 10^−6^ M. (**e**) SERS spectra of adenine adsorbed on faceted Co_3_O_4_ samples at 10^−4^ M.

**Figure 5 ijms-23-15930-f005:**
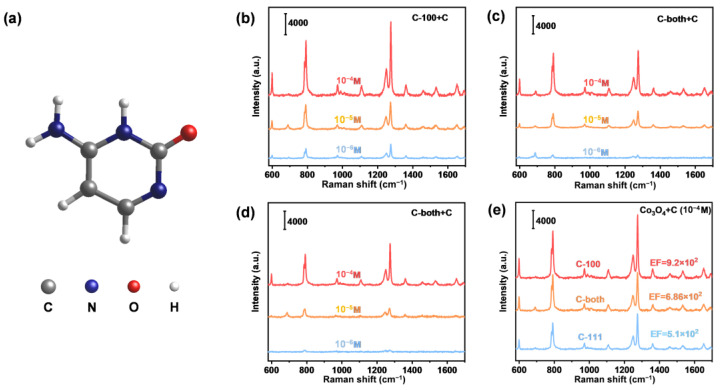
SERS spectra of cytosine adsorbed on faceted Co_3_O_4_ samples. (**a**) Molecule structure of cytosine; (**b**) SERS spectra of cytosine adsorbed on C-100 within a concentration range from 10^−4^ to 10^−6^ M; (**c**) SERS spectra of cytosine adsorbed on C-both within a concentration range from 10^−4^ to 10^−6^ M; (**d**) SERS spectra of cytosine adsorbed on C-111 within a concentration range from 10^−4^ to 10^−6^ M; (**e**) SERS spectra of cytosine adsorbed on faceted Co_3_O_4_ samples at 10^−4^ M.

**Figure 6 ijms-23-15930-f006:**
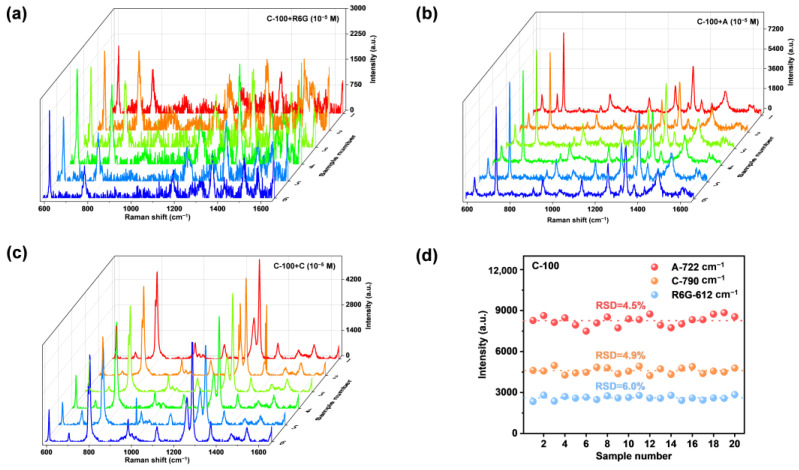
SERS spectra (**a**–**c**) and uniformity (**d**) of Raman signals of target molecules on faceted Co_3_O_4_ samples at 20 random positions.

**Figure 7 ijms-23-15930-f007:**
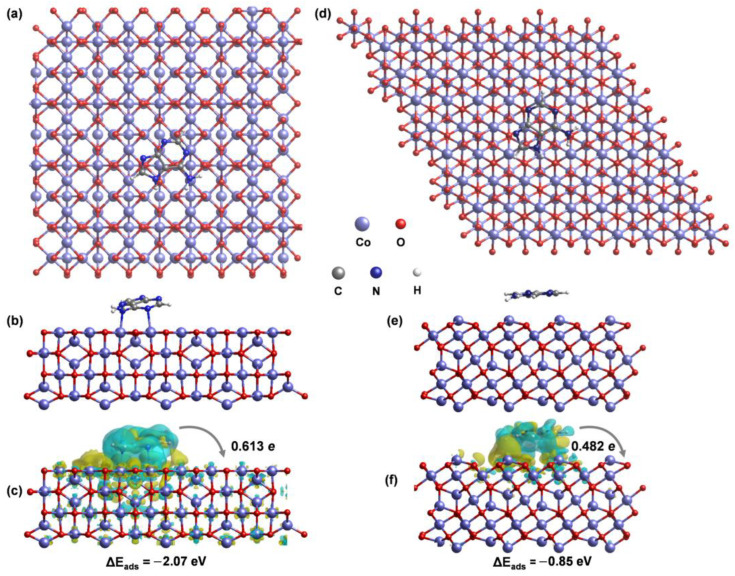
DFT calculation results. (**a**) Top view of the C-100 model; (**b**) optimized structure of adenine adsorbed on C-100; (**c**) charge density difference and adsorption energy of adenine adsorbed on C-100; (**d**) top view of the C-111 model; (**e**) optimized structure of adenine adsorbed on C-111; (**f**) charge density difference and adsorption energy of adenine adsorbed on C-111. The yellow (blue) represents the charge accumulation (depletion) regions.

## Data Availability

Data are contained within the article or Appendix A.

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
