# Peer review of "Facet-Dependent SERS Activity of Co3O4"

_ijms, 2022, doi:10.3390/ijms232415930_

Round 1

Reviewer 1 Report

In this work, the authors used three types of faceted Co3O4 microcrystals with dominantly exposed {100} facets, {111} facets, and co-exposed {100}-{111} facets as SERS substrates to detect the rhodamine 6G and nucleic acids. 

This research is of great use in the SERS field. Overall, the article is well-written with consecutiveness, strict logic, affluent datum, intensive theory, and clear consecution. Several small suggestions are supplied:

  1. Please clarify why the laser wavelength of 532 nm is used in Fig.3 and 633 nm is used in Fig. 4.
  2. In lines 181-189, the authors did not mention Fig. 7 in the text. Please check it.
  3. In line 250, please show the detailed calculation steps of EF in the text.
  4. Check the English language to eventually improve it a little bit more., e.g., Line 244 NSERS?
  5. Line 27-28, “when excitation takes place in noble and coinage metals (Au, Ag and Cu) with strongly localized surface plasmon resonances (LSPR) effect.” A reference related to the LSPR effect should be cited in the text (e.g., Nanomaterials, 2020, 10, 2030)
  6. Some references dealing with SERS should be added to improve the introduction, such as Results in Physics, 2020, 17,103168.

     So, the present manuscript is suitable for publication in this journal, subject to the above-mentioned minor revision

Author Response

We are very grateful to the referees for their careful reading of the manuscript and raising many valuable comments which are definitely helpful to the improvement of the paper quality. The responses to each question or comment are shown below.

Referee 1

In this work, the authors used three types of faceted Co3O4 microcrystals with dominantly exposed {100} facets, {111} facets, and co-exposed {100}-{111} facets as SERS substrates to detect the rhodamine 6G and nucleic acids. This research is of great use in the SERS field. Overall, the article is well-written with consecutiveness, strict logic, affluent datum, intensive theory, and clear consecution. Several small suggestions are supplied.

Response: We sincerely thank the reviewer for the positive evaluation of our work. These comments are helpful to further improve the quality of the manuscript. We supplemented a more comprehensive experiment based on these constructive comments, and made additions and corrections to the manuscript. All revisions in the manuscript and Supplementary Materials are highlighted in yellow. We believe that the revised manuscript can meet the standard of publication in International Journal of Molecular Sciences. Here, we provide point-by-point responses to the comments.

Q1: Please clarify why the laser wavelength of 532 nm is used in Fig.3 and 633 nm is used in Fig. 4.

Answer: Thanks to the reviewer for the careful review. ‘The laser wavelength of 633 nm’ is a spelling error caused by our carelessness, and we clarify all molecules were tested with a 532 nm laser.

Q2: In lines 181-189, the authors did not mention Fig. 7 in the text. Please check it.

Answer: Many thanks to this reviewer for the careful review on the manuscript. It was indeed caused by our negligence and we apologize for that. In the revised manuscript, we have supplemented it in Lines 181-191.

Density functional theory (DFT)calculations were employed to calculate the charge transfer and the adsorption behavior of target molecules. Co3O4-(100) and Co3O4-(111) models were established and investigated as substrates to adsorb adenine (Figure 7a, d). The charge density difference results of the adenine@Co3O4 models revealed that the electron transfer from adenine to Co3O4 (Figure 7b, e). The Bader charge analysis and adsorption energy calculations were employed to semi-qualitatively describe the charge transfer process and the interaction between adsorbed molecule and substrate. The C-100 (-2.07 eV) exhibited much more strong interaction with adenine than C-111 (-0.85 eV). Besides, C-100 can transfer more electrons (0.613 e) from substrate to adenine (0.482 e) (Figure 7c, f). Therefore, the facilitated CT process on {100} facets result in enhanced CM and thus a higher SERS activity than on {111} facets can be achieved.

Q3: In line 250, please show the detailed calculation steps of EF in the text.

Answer: Thanks again to the reviewer for the constructive comment on improving the manuscript. As suggested by the reviewer, we have supplemented the detailed calculation steps of EF in Lines 237-265.

To calculate the enhancement factor (EF) of Co3O4, the ratio of SERS to Normal Raman spectra (NRS) of rhodamine 6G (R6G), adenine (A) and cytosine (C) were determined by using the following calculating formula:

where ISERS and INRS refer to the peak intensities of the SERS spectrum and NRS, respectively. NSERS and NNRS refer to the number of adenine molecules in the SERS substrate and normal Raman sample, respectively. The radius of laser spot was approximately 0.5 µm, depth of laser penetration is about 1 µm. The data for R6G (bulk) on bare Si substrate was used as non-SERS-active reference. Specifically, the intensity was obtained by taking average from measurements of 20 spots, and INRS was tested to be 6027 counts. NSERS was determined from the laser spot size illuminating the sample and the density of R6G molecules adsorbed on the Co3O4 chip surface. C is the molar con-centration of the analyte solution. V is the volume of the droplet. NA is Avogadro constant. SRaman is the laser spot area (0.5 μm in radius) of Raman scanning. 20 μl of the droplet on the substrate was spread into a square of about 5mm in side length after solvent evaporation, from which the effective area of the substrate, SSub can be obtained. Regarding R6G, the density (ρ) of the R6G solid was approximately 1.15 g/cm3, the confocal depth (h) of the laser beam is 1μm, and according to molecular weight (M), NNRS was estimated to be 5×1010.

In the SERS measurements, the Raman scattering peak at 612 cm-1 was selected for the calculation of the EF. Substituting these values of the above variable into the equation, the maximum EF could be concluded to be approximately 1.12×104, while the concentration of R6G solution C=10-6 M and the Raman intensity ISERS=512 counts. The EF of adenine (5.81×103) was also calculated following this procedure and the Raman scattering peak at 722 cm-1 was selected for the calculation of the EF. INRS was tested to be 30412 counts and the Raman intensity ISERS=1336 counts. The EF of cytosine (2.78×103) was also calculated. The Raman scattering peak at 792 cm-1 was selected for the calculation of the EF. INRS was tested to be 15012 counts and the Raman intensity ISERS=315 counts.

Q4: Check the English language to eventually improve it a little bit more., e.g., Line 244 NSERS?

Answer: Thanks to the reviewer for the careful review. These were indeed caused by our carelessness, and we are sorry for that. These mistakes have been checked and corrected.

Q5: Line 27-28, “when excitation takes place in noble and coinage metals (Au, Ag and Cu) with strongly localized surface plasmon resonances (LSPR) effect.” A reference related to the LSPR effect should be cited in the text (e.g., Nanomaterials, 2020, 10, 2030).

Answer: We thank the reviewer for the valuable question. The reference mentioned above has been added to manuscript (line 28).

Q6: Some reference dealing with SERS should be added to improve the introduction, such as Results in Physics, 2020, 17,103168.

Answer: Thanks to the reviewer for the relevant literature recommended. After careful reading and research of these articles, we have added citations to these references at the appropriate places in the manuscript.

In the revised manuscript, this reference is cited (line 32).

Reviewer 2 Report

In this manuscript, the authors demonstrated that Co3O4 microcrystal with {100} facets exhibited the best SERS performance than {111} facets and {100}-{111} facets. The enhancement factor was found on the order of 10^4, which is much larger than the value expected in the chemical enhancement mechanism. I feel the manuscript meets the scope of the International Journal of Molecular Sciences, and below are some comments.

Suggestions to authors for improving the manuscript:

1. In order to compare the performance of C-100 with others, I feel using EF is more reasonable in Figures 3(e), 4(e), and 5(e).

2. When Rh6G is excited by a 532 nm laser, a fluorescence background should be observed in the Raman spectrum. Please explain why the fluorescence background is gone.

3. Is the efficiency of PICT dependent on excitation wavelength?

4. What’s the concentration of Co3O4 before being mixed with target molecules?

5. Does the EF depend on the size of the microcrystal? If so, is it possible to synthesize Co3O4 with a size of a few hundred nanometers or less?

Author Response

We are very grateful to the referees for their careful reading of the manuscript and raising many valuable comments which are definitely helpful to the improvement of the paper quality. The responses to each question or comment are shown below.

Referee 2

In this manuscript, the authors demonstrated that Co3O4 microcrystal with {100} facets exhibited the best SERS performance than {111} facets and {100}-{111} facets. The enhancement factor was found on the order of 104, which is much larger than the value expected in the chemical enhancement mechanism. I feel the manuscript meets the scope of the International Journal of Molecular Sciences, and below are some comments.

Q1: In order to compare the performance of C-100 with others, I feel using EF is more reasonable in Figures 3(e), 4(e), and 5(e).

Answer: Thanks to the reviewer for the constructive comment. Figures 3(e), 4(e), and 5(e) have been replaced by Figure R1(a), R1(b), and R1(c) in the manuscript, respectively.

Figure R1. (a) SERS spectra of R6G adsorbed on faceted Co3O4 samples at 10−4 M.(b) SERS spectra of adenine adsorbed on faceted Co3O4 samples at 10−4 M. (c) SERS spectra of cytosine adsorbed on faceted Co3O4 samples at 10−4 M.

Q2: When Rh6G is excited by a 532 nm laser, a fluorescence background should be observed in the Raman spectrum. Please explain why the fluorescence background is gone.

Answer: Thanks to the reviewer for the valuable comment. The fluorescence background of data presented in the manuscript were all removed by confocal microscopy Raman spectrometer's (LAMBDA PerkinElmer) software. The raw data were shown below (Figure R2).

Figure R2. (a) SERS spectra of R6G adsorbed on faceted Co3O4 samples. (b) SERS spectra of adenine adsorbed on faceted Co3O4 samples. (c) SERS spectra of cytosine adsorbed on faceted Co3O4 samples.

Q3: Is the efficiency of PICT dependent on excitation wavelength?

Answer: We thank the reviewer for the valuable question. Photoinduced charge-transfer (PICT) is significantly dependent on illuminate laser wavelength. As shown in Figure R3(a), adenine molecule only has a narrow absorption band around 270 nm. The laser wavelength we used in the measurement is 532 nm, 633 nm and 785 nm, which is far away from the molecule absorption band. While the inter-molecule charge transfer process was forbidden, we measured the Raman spectrum of Co3O4+A complex under different illuminate wavelength. [Chem Commun (Camb) 2018, 54 (17), 2134-2137] Figure R3(b) shows the Raman signal of Co3O4+A under 532 nm laser is much stronger than the others. The PICT under 532 nm is stronger than illuminate under 633 nm and 785 nm laser because the Raman signal enhancement originate from PICT process.

Figure R3. (a) UV-Vis absorption spectrum of adenine water solution. (b) Raman signal of Co3O4+A under 532 nm 633 nm and 785 nm laser.

Q4: What’s the concentration of Co3O4 before being mixed with target molecules?

Answer: We thank the reviewer for the valuable question. Co3O4 is a black solid powder and insoluble in water, 5 mg powder was evenly dispersed in 2 mL solution of test substance for testing.

Q5: Does the EF depend on the size of the microcrystal? If so, is it possible to synthesize Co3O4 with a size of a few hundred nanometers or less?

Answer: Thanks to the reviewer for the valuable comment. According to the previous research, chemical mechanism (CM) dominates the enhancement of Co3O4 materials due to the lack of surface plasmon resonance (SPR) effect. Only a few kinds of semi-metallic semiconductors show size dependence on SERS. [Cell Reports Physical Science 2020, 1 (3); Nat Commun 2020, 11 (1), 3889] In our case, {100} exposed Co3O4 does not show any semi-metal properties so there’s no need to optimize the size as a critical parameter.

Reviewer 3 Report

In this contribution the authors provided mechanistic insights into the facet-dependent SERS activity of Co3O4(100)/(111) both experimentally and computationally. While the study was well carried out and it has a nice impact in the field, there are a few concerns regarding the computational details:

1.     The authors attempted to explain the SERS mechanism using adsorption energy and Bader charge. Would these two be sufficient? The reviewer believes there should be at least some Raman spectra calculations as well.

2.     How were the (100) and (111) surfaces determined? Do they have the lowest surface energy among different surface terminations?  

3.     It is well known that for molecule adsorption, PBE-D3 (or D2) should be considered. The authors should also test RPBE-D3 as well.

4.     For Co oxides, have the authors considered DFT+U?

5.     For these model surfaces 1x1x1 k-point should be okay; it would be better to provide the results for 2x2x1 just to double check.

6.     How were the slabs relaxed? Did the authors fix the bottom layer, or all atoms were allowed to relax? What were the convergence criteria? All of these need to be specified in the method section to make sure the results can be reproduced. If possible, all INCAR and CONTCAR should be provided as supplementary materials.

7.     Ref[47] is not needed.

Author Response

We are very grateful to the referees for their careful reading of the manuscript and raising many valuable comments which are definitely helpful to the improvement of the paper quality. The responses to each question or comment are shown below.

Referee 3

In this contribution the authors provided mechanistic insights into the facet-dependent SERS activity of Co3O4(100)/(111) both experimentally and computationally. While the study was well carried out and it has a nice impact in the field, there are a few concerns regarding the computational details:

Q1The authors attempted to explain the SERS mechanism using adsorption energy and Bader charge. Would these two be sufficient? The reviewer believes there should be at least some Raman spectra calculations as well.

Answer: We truly thank the reviewer for this good suggestion. The charge density difference and Bader charge analysis have been proved as efficient tools to support and clarify interfacial charge transfer (CT) process between adsorbed molecules and the SERS substrate. [Small 2018, 14 (8), 1703274; ACS Applied Materials & Interfaces 2021, 13 (22), 26551-26560; Biosensors and Bioelectronics 2021, 191, 113452] High charge transfer efficiency accounts most for improved chemical enhancement mechanism. [The Journal of Physical Chemistry Letters 2014, 5 (6), 964-968; The Journal of Physical Chemistry C 2015, 119 (40), 23113-23118]. Raman spectra calculations can be conducted to predict the conformation and adsorption sites of analyte on SERS substrate. In principle, through comparison between the theoretical computation and the experimental SERS spectra, the adsorption of molecule on the surface of the metal can be analyzed and the adsorption conformation can be determined. [The Journal of Physical Chemistry C 2018, 122 (27), 15241-15251] The focus of the present work was to demonstrate a facet-dependent SERS phenomenon on Co3O4 microcrystals. DFT calculations were here carried out to help understand the involved charge transfer process based on the most probable configurations. Since the adsorption conformation of adenine on the Co3O4 surface has not been solved, theory and experiment can be combined to decide the real corresponding conformations in more and more promising future works.

Q2How were the (100) and (111) surfaces determined? Do they have the lowest surface energy among different surface terminations?  

Answer:  Thank you for this question. The surfaces were selected based on experimental results (Figure R4). The results are consistent with previously reported literature. [Advanced Materials 2012, 24 (42), 5762-5766.] According to literature, the surface energy of Co3O4 follow the order: {111}>{112}>{110}>{100}, and their corresponding values are 2.31, 1.46, 1.31 and 0.92 J m-2, respectively. [Scientific Reports 2014, 4 (1), 5767]

Figure R4. Microstructures of different faceted Co3O4 samples.

Q3It is well known that for molecule adsorption, PBE-D3 (or D2) should be considered. The authors should also test RPBE-D3 as well.

Answer:  Thanks for your reminding. Our calculations of adenine adsorption on Co3O4 were performed using the density functional theory (DFT-D3(BJ), ivdw=12) to correct for the van der Waals interaction. We have added the details into revised manuscript. The RPBE-D3 and PBE-D3 results of adsorption energy are listed below (Table R1). Similar trend was obtained.

Table R1. The RPBE-D3 and PBE-D3 results of the adsorption energies of Co3O4-100 and Co3O4-111.

Adsorption Energy                              Co3O4-100                             Co3O4-111

PBE-D3 (eV)                                           -2.07                                       -0.85

RPBE-D3 (eV)                                        -1.46                                       -0.53

Q4For Co oxides, have the authors considered DFT+U?

Answer:  Thanks for your kind reminding. U-J=5.9 eV was adopted in the whole calculations according to previous literature. [Physical Review B 2011, 83 (24), 245204] We have added the details into revised manuscript.

Q5:  For these model surfaces 1x1x1 k-point should be okay; it would be better to provide the results for 2x2x1 just to double check.

Answer:  Thanks for your suggestion. We have checked three models using 2x2x1 k-point. The differences are listed below (Table R2).

Table R2. The derived TOTEN values (energy/eV) of three models based on different k-points.

Energy                                          1x1x1                                        2x2x1

Co3O4-100                                 -3051.65551428                       -3051.57551137

Adenine                                      -104.86483445                         -104.86366519

Adenine@Co3O4-100                 -3158.59230215                      -3158.50275931

The minor differences can ensure the same trend demonstrated in the original results using 1x1x1 k-point.

Q6:  How were the slabs relaxed? Did the authors fix the bottom layer, or all atoms were allowed to relax? What were the convergence criteria? All of these need to be specified in the method section to make sure the results can be reproduced. If possible, all INCAR and CONTCAR should be provided as supplementary materials.

Answer:  Thanks for your suggestion. The bottom six layers of the Co3O4-100 and ten layers of the Co3O4-111 slabs were fixed during the relaxation, whereas the top two layers were fully relaxed until the energy and force converged within 10−4 eV and 0.02 eV Å−1, respectively. The INCAR and CONTCAR for geometric optimization of adenine@Co3O4-100 and adenine@Co3O4-111 are provided as supplementary materials now.

Q7:  Ref [47] is not needed.

Answer:  Thanks for your suggestion. The previous Ref [47] has now been removed in the revised manuscript.

Round 2

Reviewer 3 Report

Most of the questions have been carefully addressed.  The reviewer appreciates the efforts by the authors.

In Q2, the reviewer was wondering the surface terminations of each facet: e.g. Co-terminated (111) vs. O-terminated (111).  If the authors have tested this, it would be good to point out in the Method section.

Author Response

Most of the questions have been carefully addressed.  The reviewer appreciates the efforts by the authors.

Response: We sincerely thank the reviewer for the positive evaluation of our work.

In Q2, the reviewer was wondering the surface terminations of each facet: e.g. Co-terminated (111) vs. O-terminated (111).  If the authors have tested this, it would be good to point out in the Method section.

Answer:  Thank you for this question.

Figure R1. Possible exposed surface configurations of (a) Co3O4-100 and (b) Co3O4-111.

As illustrated in Figure R1. We considered 4 types of configurations with different terminates for Co3O4-100 and 3 types for Co3O4-111.As shown in Figure ref, the final calculated slab models as the reviewer mentioned (Co3O4-111) are consistent with literature. [Journal of Materials Chemistry A 2022, 10 (20), 10837-10843]. Honestly, our optimized Co3O4-100 does not agree with that paper. We have added corresponding description in “DFT calculation section “.

Figure ref. Models of (100) and (111) exposed Co3O4 from literature.